# Halo-fluorescein for photodynamic bacteria inactivation in extremely acidic conditions

Ying Wang[1,2], Jiazhuo Li[1,2], Zhiwei Zhou[3], Ronghui Zhou[4], Qun Sun[3] & Peng Wu [1,2✉]

Aciduric bacteria that can survive in extremely acidic conditions (pH < 4.0) are challenging to the current antimicrobial approaches, including antibiotics and photodynamic bacteria inactivation (PDI). Here, we communicate a photosensitizer design concept of halogenation of fluorescein for extremely acidic PDI. Upon halogenation, the well-known spirocyclization that controls the absorption of fluorescein shifts to the acidic pH range. Meanwhile, the heavy atom effect of halogens boosts the generation of singlet oxygen. Accordingly, several photosensitizers that could work at even pH < 2.0 were discovered for a broad band of aciduric bacteria families, with half maximal inhibitory concentrations (IC$_{50}$) lower than 1.1 μM. Since one of the discovered photosensitizers is an FDA-approved food additive (2′,4′,5′,7′-tetra-iodofluorescein, TIF), successful bacteria growth inhibition in acidic beverages was demonstrated, with greatly extended shelf life from 2 days to ~15 days. Besides, the in vivo PDI of Candidiasis with TIF under extremely acidic condition was also demonstrated.

[1] State Key Laboratory of Hydraulics and Mountain River Engineering, Sichuan University, 610064 Chengdu, China. [2] Analytical & Testing Center, Sichuan University, 610064 Chengdu, China. [3] College of Life Science, Sichuan University, 610064 Chengdu, China. [4] State Key Laboratory of Oral Diseases, West China Hospital of Stomatology, Sichuan University, 610041 Chengdu, China. ✉email: wupeng@scu.edu.cn

The compact between humans and pathogenic micro-organisms has been lasting for thousands of years. From mild skin infections to deadly ulcers and poisoning, bacteria threaten human health all the time[1,2]. The activity of bacteria is related with pH, and for most microorganisms, the livable pH is 5.0–9.0 (ref. [3]). However, a great number of bacteria, namely, aciduric bacteria, can survive in extremely acidic conditions (pH < 4.0, e.g., *Helicobacter pylori*, pH < 2.0)[3–5]. In such pH range, the effects of common antibacterial agents (such as antibiotics and phage) would be weakened or even lost (Fig. 1a). For example, penicillin, a well-known antibiotics, would be acidified into penicillenic acid at pH 4 and lost its antibiotics effect (Supplementary Fig. 1)[6–8]. Therefore, it is a great challenge to develop antimicrobial agents that can work in extremely acidic conditions.

Photodynamic inactivation (PDI) is a promising method to kill pathogenic microorganism. During such a process, photosensitizers (PSs) sensitize ground state oxygen ($^3O_2$) into singlet oxygen ($^1O_2$) with strong oxidation capacity, which can damage and even kill bacteria[9,10]. Due to its destructive nature, the drug resistance of bacteria PDI agents is largely lower than that of antibiotics[11]. However, most PSs contain easily protonated sites (e.g., N and O), which will lose the photodynamic activity under extremely acidic condition (Supplementary Fig. 1).

To facilitate PDI in extremely acidic conditions (pH < 4.0), a PS design concept was proposed here, based on the well-known pH-responsive intramolecular spirocyclization reaction of the fluorescein derivatives[12,13] and the heavy atom effect (HAE) of halogens. Fluorescein and its derivatives bear an intramolecular nucleophile equilibrium between a colored open form and a colorless spirocyclic form in solution (Fig. 1b and Supplementary Fig. 2)[14,15], which has been widely used for design of various fluorescent probes[16–20]. Upon halogenation, the electron withdrawing nature of Cl, Br, and I will lower the pH threshold of the spirocyclization reaction. On the other hand, grafting fluorescein

with heavy atoms (Cl, Br, and I) will boost the singlet oxygen generation due to the well-known HAE. Therefore, halogenation of the fluorescein derivatives are expected to generate PSs suitable for PDI in extremely acidic conditions.

On the basis of the above design, here we discovered several PSs that could work at even pH < 2.0. Most importantly, one of the PSs here, 2′,4′,5′,7′-tetraiodofluorescein (TIF), is an FDA (Food and Drug Administration agency)-approved food additive[21], which therefore permits photodynamic antimicrobial applications for edible acidic foods. Moreover, the in vivo photodynamic antimicrobial therapy (PACT) of oral Candidiasis with TIF under extremely acidic condition was successfully demonstrated.

## Results

**Evaluation of the intramolecular spirocyclization of the fluorescein derivatives.** To develop PS that can work in extremely acidic media, we considered that PS should meet two criteria: efficient light absorption and high singlet oxygen quantum yield in such condition. On the basis of the pH responsiveness of fluorescein (Fig. 2a and Supplementary Fig. 2), we envisioned that grafting halogen atoms to yield halo-fluorescein may be effective. On one hand, the intramolecular spirocyclization reaction is influenced by the electron density of the core structure of fluorescein[17,22,23], and the electronegativity of halogens is expected to be helpful for shifting the reaction to low pH range, resulting in efficient absorption in such range. On the other hand, halogens are the most popular choices of HAE, which can promote the generation of singlet oxygen ($^1O_2$) via increasing the probability of intersystem crossing[24]. To optimize these parameters, a series of halogen-substituted fluorescein derivatives were investigated here, through replacement with Cl, Br, and I on both of the xanthene and the benzene moieties. These PSs can be categorized into four groups (detailed chemical structures and

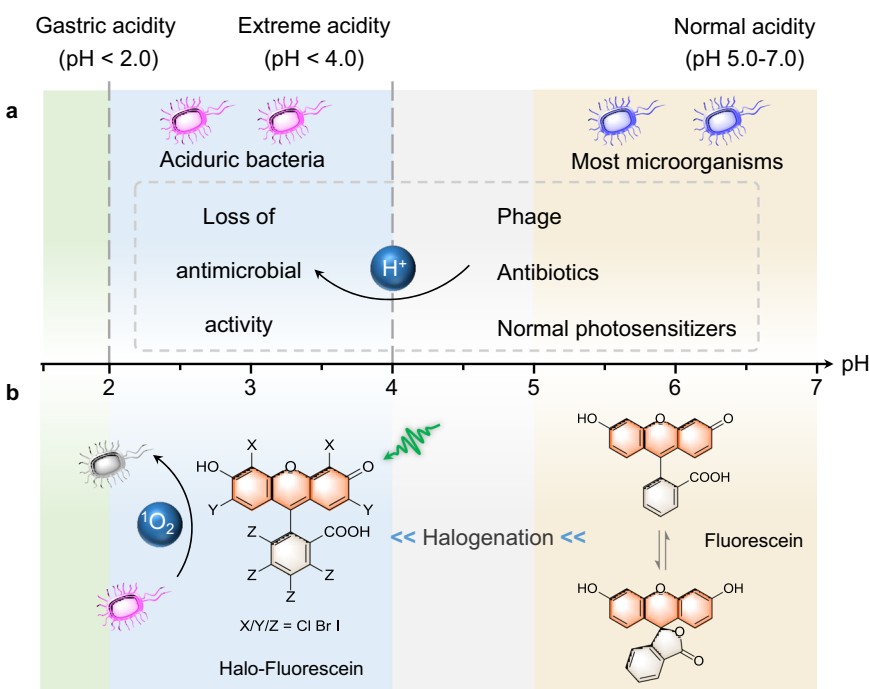

**Fig. 1 Schematic illustration of halo-fluorescein generation singlet oxygen ($^1O_2$) in extremely acidic condition. a** Survival pH of most microorganisms and aciduric bacteria. In extremely acidic media (pH < 4.0), phage, antibiotics, and normal photosensitizers will lose their antibacterial activity; and **b** halogenated fluorescein exists in open form under extremely acidic condition, which can generate singlet oxygen upon photo irradiation for bacteria inactivation.

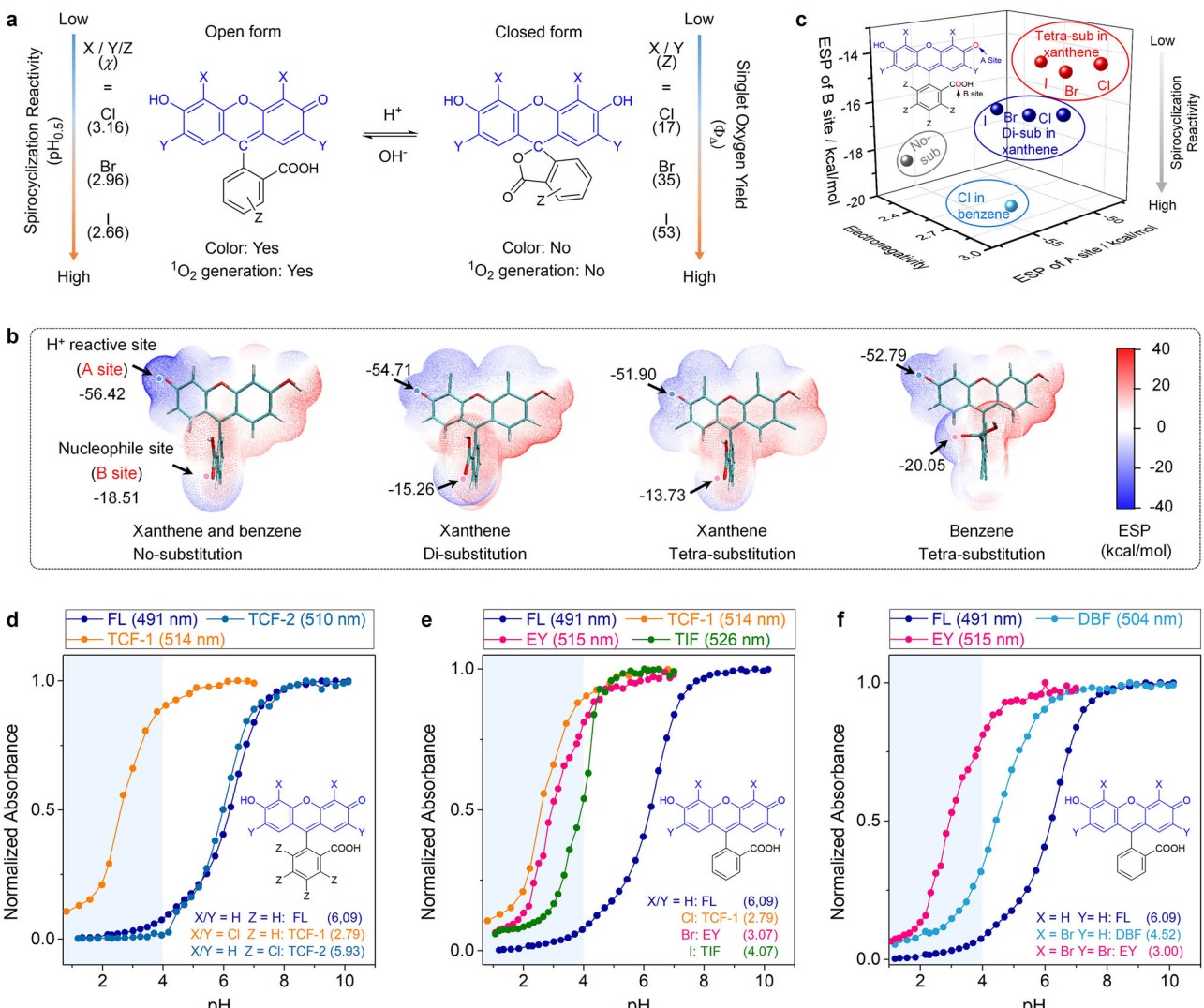

**Fig. 2 Evaluation of the intramolecular spirocyclization of fluorescein derivatives. a** Thermal equilibrium of intramolecular spirocyclization between the fluorescent open form and the nonfluorescent closed form, here χ and Z represents electronegativity and atomic number of chlorine, bromine, and iodine, respectively; **b** the electrostatic potential (ESP) map of FL (no-substitution), DCF (di-substitution in xanthene), TCF-1 (tetra-substitution in xanthene), and TCF-2 (tetra-substitution in benzene); **c** the relationship between electronegativity and ESP of deprotonated hydroxyl group on the xanthene core (site A) and the carboxyl group on the benzoic acid (site B), including non-substituted group (gray), chloro-substituted in benzene moiety (cyan), di-substituted in xanthene moiety (blue), and tetra-substituted in xanthene moiety (red ball); **d** pH titration results of the fluorescein derivatives with different substituting position; **e** pH titration results of the fluorescein derivatives with different kinds of halogen substitutions; and **f** pH titration results of the fluorescein derivatives with different number of halogen substitutions.

names are given in Fig. 3): chloro-substitution in the benzene moiety (TCF-2), di-substitution (DCF, DBF, and DIF), and tetra-substitution (TCF-1, EY, and TIF) in the xanthene ring, and octa-substitution (OCF, PB, and RB).

To estimate the influence of halogen substitution on the spirocyclization reaction of fluorescein, we first carried out electrostatic analysis[25]. As shown in the electrostatic potential map (Fig. 2b and Supplementary Fig. 5), the neutral form of fluorescein has two negative charge sites, namely, the deprotonated hydroxyl group on the xanthene ring (site A, as a $H^+$ reactive site) and the carbonyl in the benzene ring (site B, as a nucleophile site). Besides, increasing the negative charge density of A and B sites will promote the spirocyclization reaction. Upon substitution on the xanthene ring, halogen atoms are electron withdrawing for both site A and site B, thus decreasing the electron density and inhibiting the formation of the closed-ring form. Besides, tetra-substitution contributes to lower reaction

activity over di-substitution. Instead, substitution on the benzene ring results in withdrawing for site A, but electron donating for site B. Therefore, halogenation of benzene contributes slighter than that of xanthene ring on the spirocyclization reaction. HOMO and LUMO analysis also indicated that their electron clouds are mainly distributed on the xanthene ring, and introducing of halogens leads to shifting of the electron cloud, i.e., the reaction activity of the xanthene ring (Supplementary Table 3). Accordingly, the spirocyclization reaction activity is determined by the electronegativity of halogen atoms, the number, and position of their substitution, with lowest activity from tetra-substituted fluorescein (Fig. 2c).

The $pH_{0.5}$ values (the pH value when the open form reduce to the half of the maximum) of various fluorescein derivatives were determined experimentally through pH titration (Supplementary Figs. 6–16). Generally, the $pH_{0.5}$ values are lowered upon halo-substitution (Fig. 3). Specifically, the $pH_{0.5}$ values of TCF-1 (2.79,

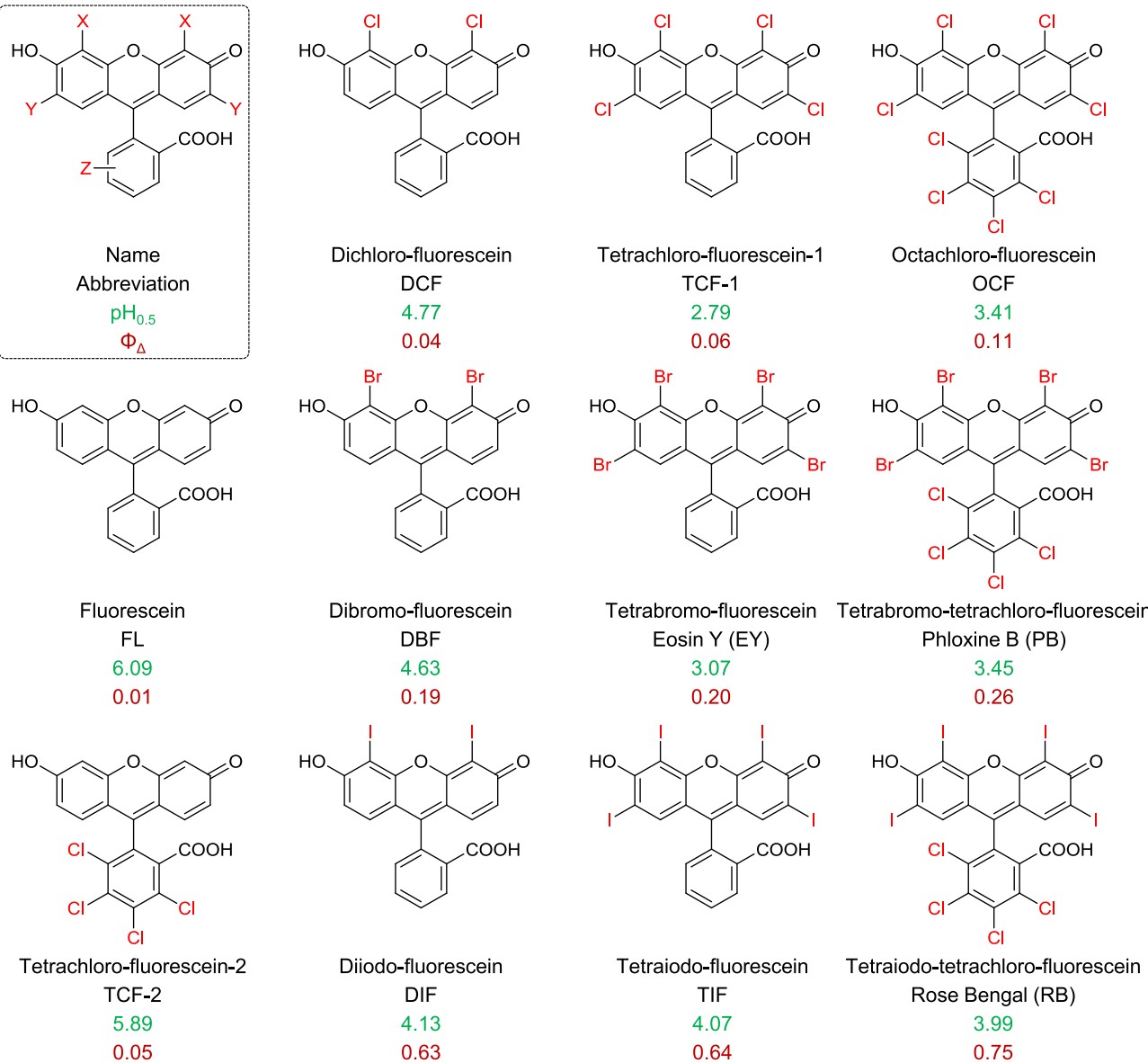

**Fig. 3 The pH$_{0.5}$ and singlet oxygen quantum yield ($\Phi_\Delta$) of the fluorescein derivatives in this work.** For the measurement of $\Phi_\Delta$, pH = 10 media was employed.

chlorine-substitution on xanthene) and TCF-2 (5.93, chlorine-substitution on benzene) are lower than fluorescein (6.09) by 3.30 and 0.16 units (Fig. 2d), respectively, indicating that substitution on xanthene is more pronounced than that on benzene. Meanwhile, the pH$_{0.5}$ values of TCF-1, EY, and TIF are lowered by 3.30, 3.02, and 2.02 units as compared with fluorescein (Fig. 2e). Furthermore, the pH$_{0.5}$ values are also dependent on the number of the substituents (Fig. 2f). For example, tetra-bromine fluorescein derivatives (EY) exhibits a lower pH$_{0.5}$ value (3.07) than di-bromine fluorescein (DBF, 4.52). Overall, by introducing of electron withdrawing halogens onto the skeleton of fluorescein, the pH$_{0.5}$ values were successfully lowered to the strongly acidic range (pH < 4, Fig. 3), which agreed well with the theoretical electrostatic analysis. Therefore, halo-substitution permits partial open form of the fluorescein derivatives in extremely acidic conditions, resulting in the efficient absorption.

Besides down-shifting the pH of the spirocyclization reaction, halogen substitution also results in substantial increase of singlet oxygen generation. To permit the open-form structure of all fluorescein derivatives, all the measurements were carried out in pH = 10 media. As shown in Fig. 3, the relative singlet oxygen quantum yields ($\Phi_\Delta$, with RB as the standard[26]) of the fluorescein derivatives generally follow the order of I > Br > Cl, with maximum $\Phi_\Delta$ of 0.75 from RB (tetraiodo-xanthene and tetrachloro-benzene). Besides, substitutions on the xanthene moiety are more effective for boosting the singlet oxygen generation than that on benzene. Probably, the electron clouds are concentrated on the xanthene moiety, which receive greater spin–orbit coupling effect by halogen atom (Supplementary Table 3)[27,28]. Related photophysical studies (section S6 in Supplementary Information, Supplementary Figs. 20–45, and Supplementary Table 4) confirm the existence of HAE in these halo-substituted fluorescein derivatives.

**pH-dependent singlet oxygen generation from TIF.** To evaluate the pH-dependent $^1O_2$ generation, TIF, PB, and RB were chosen here because of their high $^1O_2$ generation efficiency and low pH$_{0.5}$. As shown in Fig. 4a, with the decrease of pH, the characterized $^1O_2$ phosphorescence emission (1270 nm) from TIF was gradually decreased, which was in agreement with the above pH

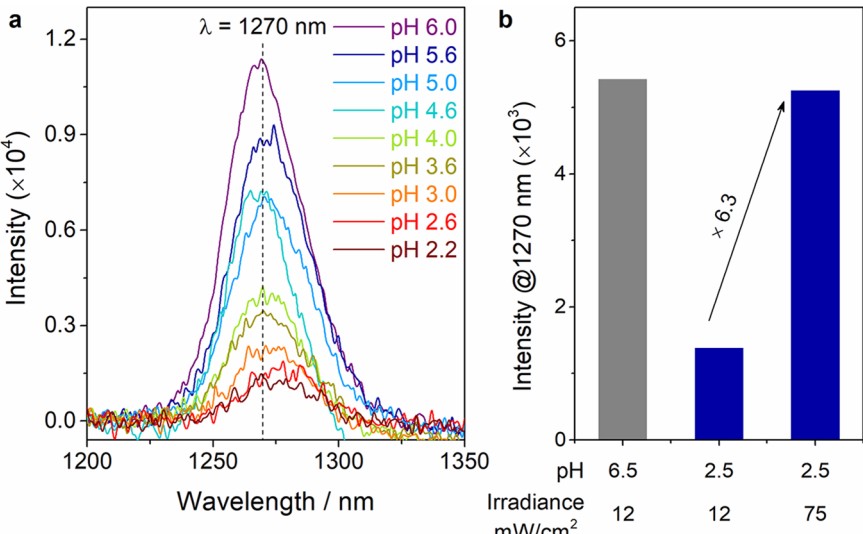

**Fig. 4 Solution pH-dependent 1O2 generation of TIF. a** The $^1O_2$ phosphorescence emission of TIF under different pH; and **b** the $^1O_2$ phosphorescence emission intensity of TIF under pH 6.5 or pH 2.5, followed by different light irradiance.

titration results. Similar results from PB and RB were also observed (Supplementary Figs. 17 and 18). When pH was lowered down to 2.2 (the gastric acidity), appreciable $^1O_2$ phosphorescence emission was still collected from TIF, indicating the good performance of TIF in photosensitized generation of $^1O_2$ in extremely acidic condition. Besides, the pH-induced decrease of $^1O_2$ generation could be compensated through enlarging the light irradiance. As shown in Fig. 4b, when increasing the light irradiance from 12 to 75 mW/cm$^2$, the intensity of $^1O_2$ at pH 2.5 was almost the same as the intensity at pH 6.5 (Supplementary Fig. 19). Therefore, TIF featured good performance for potential photodynamic bacteria inactivation at extremely acidic conditions.

**Light-introduced bacteria killing in extremely acidic conditions.** On the basis of the excellent $^1O_2$ generation performance of TIF, PB, and RB in extremely acidic conditions, their photodynamic sterilization performances were evaluated with *Lactobacillus plantarum* (*L. plantarum*). *L. plantarum* is a typical bacteria that can survive in media of pH < 4.0 (ref. [29]). For comparison, two well-known PSs used in clinical applications, namely Zinc(II) tetraphenylporphyrin (Zn-tpp) and phthalocyanine (PC)[30], were also included. As shown in Fig. 5a, in the presence of the above PSs (0.5 μM) and light irradiation (LED, 3 V, 3 W), the survival rates of *L. plantarum* followed the order of TIF < RB < PB < PC < Zn-tpp (Supplementary Fig. 46). For Zn-tpp, no obvious antimicrobial activity was observed. Therefore, TIF, PB, and RB could be employed as potential candidate PSs for extremely acidic photodynamic sterilization. Importantly, among these PSs, TIF is a food additive approved by FDA (Code of Federal Regulations, Title 21, Part 74.303), European Union (European Parliament and Council Directive 94/36/EC | ANNEX III), and Chinese National Standard (GB 2760-2011; Supplementary Table 5). Since pathogenic and spoilage bacteria can grow in some acidic foods (e.g., fruit juices of pH < 3), and thus cause great health threatening and also serious economic losses[31,32], photodynamic sterilization of extremely acidic foods with TIF is thus expected to be appealing.

As shown in Fig. 5b, considerable antimicrobial activity of TIF was received in both of the neutral and acidic pH range (2.5–7.0, Fig. 5b and Supplementary Fig. 47). Further dark toxicity investigations indicated that at TIF concentration up to 50 μM

(similar to the maximum level in food guided by Chinese National Standard GB 2760-2011, 57 μM), no significant inhibition of bacteria growth was observed (Fig. 5c). Through EPR and scavenger study (Supplementary Figs. 48 and 49), the reactive oxygen species during bacteria killing were identified as singlet oxygen ($^1O_2$) and superoxide anion ($\cdot O_2^-$), which were produced from type-II and type-I photosensitization processes[33], respectively. Therefore, the excellent bacteria killing performance of TIF could be ascribed to the photodynamic effect.

The interaction of TIF with *L. plantarum* was studied from the aspects of charge interactions. Fluorescein derivatives have three protonating sites, which can be converted from the dianion form to the neutral form and eventually the monocation form under acidic conditions (Supplementary Fig. 2)[34]. Upon acidification, TIF will be stepwisely protonated, which reduces the electrostatic repulsion between TIF and negatively charged cell membrane. As shown in Fig. 5d, the surface of *L. plantarum* was negatively charged and decreased upon acidification (pH 7.4 versus pH 2.5, PBS buffer). Although the charge of TIF is less negative than that of *L. plantarum*, the overall negative charge was somewhat increased after mixing of *L. plantarum* with TIF, demonstrating the interaction between TIF and the cell membrane of *L. plantarum*. Besides, at pH 2.5, even redder precipitation of *L. plantarum* + TIF was observed as compared to pH 7.4 (inset in Fig. 5d), confirming better affinity upon acidification. Therefore, acidification of the media added affinity between TIF and *L. plantarum*, which is advantageous for the ROS to attack the bacteria[35]. The surface charge of *Escherichia coli* (*E. coli*, G$^-$) and TIF in different solvents also confirmed such interaction (Supplementary Fig. 50).

The phototoxicity of TIF for *L. plantarum* killing was further confirmed through live/dead bacterial staining with the SYTO 9/propidium iodide kit. After staining, live and dead bacterial will exhibit green and red fluorescence, respectively. As shown in Fig. 5e, without PS TIF, the red fluorescence was very weak. In contrast, bright red fluorescence was observed after introducing of TIF. Meanwhile, the membrane integrity of the treated bacteria was investigated with scanning electron microscopy (SEM) and transmission electron microscopy (TEM). After treating with TIF and light, obvious deformation and distortion parts were found on the bacteria membrane (red arrows in Fig. 5f), together with lots of lesions and holes in the intracellular components of bacteria (yellow arrows in Fig. 5g). But for the control groups

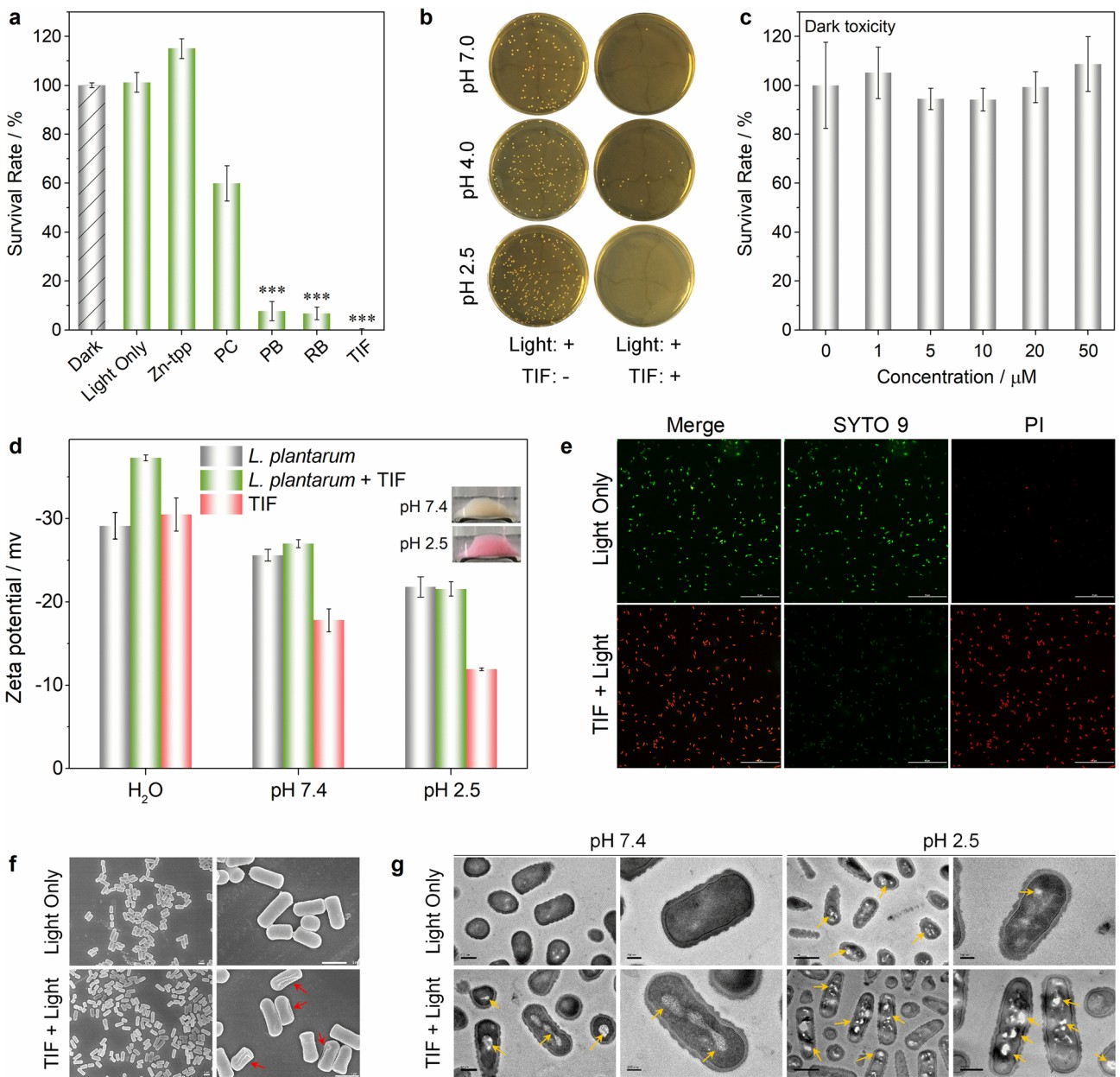

**Fig. 5 Evaluation of the photodynamic bacteria inactivation performance of TIF (light source: 520 nm green LED, 35 mW/cm$^2$) with *L. plantarum* as a model bacteria in extremely acidic conditions. a** Survival rates of *L. plantarum* after treated with 0.5 µM photosensitizers and irradiated for 10 min (pH = 2.5, coating plate method, Supplementary Fig. 46; for Zn-tpp and PC, 620 nm red LED was used; *$P < 0.05$, **$P < 0.01$, ***$P < 0.001$). Error bars represent standard deviation, $n = 3$ independent replicates; **b** pictures of the bacterial cultures under different conditions (bacteria were first treated and then plated on solid media for colony quantification); **c** dark toxicity evaluation of TIF. Error bars represent standard deviation, $n = 3$ independent replicates; **d** Zeta potentials of *L. plantarum*, *L. plantarum* + TIF, and TIF in different media (inset: the pictures of *L. plantarum* + TIF at different pH, washed twice to remove unabsorbed TIF). Error bars represent standard deviation, $n = 3$ independent replicates; **e** live/dead bacteria staining of TIF-treated *L. plantarum* (scale bar: 20 µm; pH 2.5; concentration of TIF: 10 µM; irradiation time: 1 h); **f** SEM images of *L. plantarum* with or without TIF under light irradiation (scale bar: 1 µm; the same condition as **e**); and **g** TEM images of *L. plantarum* with or without TIF under light irradiation (scale bar: 0.5 µm and 200 nm; the same condition as **e**).

(without TIF), no appreciable morphological change was observed. Therefore, the decreased survival rates of *L. plantarum* could be ascribed to photodynamic killing of bacteria mediated by TIF at extremely acidic conditions.

**Photodynamic inactivation of aciduric bacteria with TIF**. Next, the viability of TIF for photodynamic sterilization in extremely acidic conditions was demonstrated with broad microscopic aciduric bacteria families, including Gram-positive (G$^+$) *L. plantarum*, *Alicyclobacillus acidoterrestris* (*A. acidoterrestris*),

*Staphylococcus aureus* (*S. aureus*), *and* methicillin-resistant *S. aureus* (MRSA), Gram-negative (G$^-$) *E. coli*, *Salmonella enterica* (*Salmonella*), and *H. pylori*, and fungus *Candida albicans* (*C. albicans*). All these bacteria can survive in extremely acidic conditions, and some of these bacteria are classified as Risk Group 2 biohazardous agents by NIH[36]. For quantitative comparison, the half-maximal inhibitory concentrations (IC$_{50}$) of TIF for these bacterial were measured (Supplementary Table 6, and Supplementary Figs. 51–58). As shown in Fig. 6, excellent antimicrobial activity of TIF under extremely acidic conditions for all the

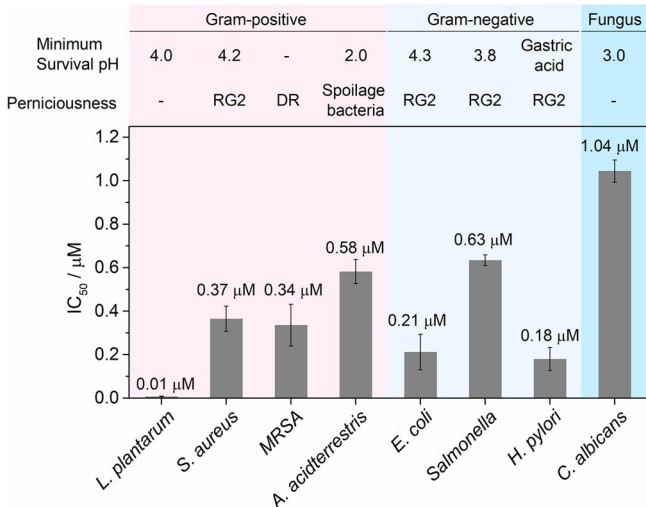

**Fig. 6 IC$_{50}$ of TIF for *L. plantarum*, *S. aureus*, MRSA, *A. acidoterrestris*, *E. coli*, *Salmonella*, *H. pylori*, and *C. albicans*.** Experimental conditions: pH 2.5 (pH 3.0 for *C. albicans*); LED irradiation, 520 nm, 35 mW/cm². RG2 Risk Group 2 (NIH Guidelines); DR drug resistant. Error bars represent standard deviation, $n = 6$ independent replicates.

investigated bacteria was received, with IC$_{50}$ values far lower than the maximum level in food guided by Chinese National Standard GB 2760-2011 (57 μM). Compared with the reported PSs, the antimicrobial performance of TIF was similar or even better (Supplementary Table 7), but under extremely acidic conditions (pH 2.5). Interestingly, the antimicrobial performances of TIF toward *S. aureus* (IC$_{50}$ of 0.37 μM) and MRSA (a well-known antibiotic-resistant bacterium listed on the high-level global priority list of WHO[37], IC$_{50}$ of 0.34 μM) were almost the same, confirming that TIF-based PDI could overcome the bacteria drug resistance (even at the extremely acidic conditions). Particularly for *H. pylori* that can survive in highly acidic gastric media (pH < 2.0), intriguing antimicrobial activity of TIF was still observed (IC$_{50}$ of 0.18 μM). Moreover, for the acid- and heat-resistant *A. acidoterrestris*, excellent photodynamic sterilization activity of TIF was also received (IC$_{50}$ of 0.58 μM). Therefore, it is clear that TIF exhibited excellent photodynamic sterilization performances for a broad band of bacteria in extremely acidic conditions.

**Photodynamic sterilization of acidic juices with TIF.** Fruit juices are public favorite acidic beverages with both good tastes and high nutritional values (Supplementary Table 8). However, the contents of the juices are also favored for bacteria growth, such as *E. coil, Salmonella, S. aureus, A. acidoterrestris,* and *C. albicans*. For storage and freshness preservation, most of the juices are sterilized by high pressure and ultrasound processing during industrial production[38–40], which is expensive and not popularized for publics. Meanwhile, lots of preservatives, such as sodium benzoate and potassium sorbate, are often added. Although the FDA agencies of various counties have legislated the maximum permitted levels for the food preservatives, profit-driven illegal and excessive usage are often reported. Therefore, developing low cost and public-friendly sterilization methods are highly desirable. Since the PS TIF here is an approved food additive by worldwide FDA agencies, we thus explored its sterilization performance in acidic fruit juices.

First, the biocompatibility of TIF was assessed with a normal cell line (L929 mouse fibroblasts) through a standard Cell Counting Kit-8 assay. As shown in Supplementary Fig. 59, upon incubating the L929 cells with various concentrations of TIF (0–20 μM) for 24 h, the survival rates were close to 100%,

indicating low toxicity of TIF. Subsequently, the lighting wavelength-dependent $^1O_2$ generation from TIF was investigated to explore the potential of TIF using continuous light (such as white LED and sunlight). The fluorescence intensity of SOSG (singlet oxygen sensor green, a fluorescence probe for $^1O_2$) with TIF (5 μM) under different wavelengths of LED for 2 min were collected. As shown in Supplementary Fig. 60, under the irradiation of purple, blue, cyan, green, and white light, the fluorescence intensity of SOSG increased obviously, indicating that TIF (maximum absorption: 526 nm) possessed the potential to utilize continuous light in the range of 400–600 nm for photodynamic sterilization (Supplementary Fig. 61). Here, a xenon lamp equipped with a 420 nm long-pass filter was chosen as the light source for subsequent studies, the power density (100 mW/cm²) of which may also permit higher irradiance over green LED at a specific penetration depth.

To investigate the potential of TIF for photodynamic sterilization in acidic beverages, five different fruit juices (Fig. 7a), including fresh passionfruit lemon (pH 2.8), tomato (pH 4.6), grape (pH 4.3), commercial grapefruit (pH 3.6), and commercial grape (pH 3.6), were chosen here. In acidic fruit juices, majority of TIF existed in the colorless closed form and thus would not affect the color of these beverages. Besides, after 10–20 min of light irradiation, all the juices had negligible change in texture, color, and fragrance (Fig. 7a). Therefore, the introduction of TIF and later light treatment would not add appreciable visual burden to the customers that consume these beverages.

To evaluate the photodynamic sterilization performance of TIF, freshly prepared and commercially available fruit juices were added with 10 μM TIF (fivefold lower than the maximum level in food guided by Chinese National Standard GB 2760-2011) and then placed into headspace bottles (Fig. 7b). The prepared juice samples were divided into two groups: one for light treatment of 10–20 min (xenon lamp with 420 nm long-pass filter, 100 mW/cm²), and another without (stored at 4 °C). The total number of bacteria colonies at different time intervals was measured based on the method suggested by Chinese National Standard (GB 4789.2-2016). As can be seen from Fig. 7c, after photo-treatment for 10 min, the shelf life of passionfruit lemon juice (pH = 2.8) could be extended to 15 days, while bacteria grew rapidly in the control group (Supplementary Fig. 62). For other juices (pH 3.6–4.6), similar bacteria inhibition results were obtained (Supplementary Figs. 63–66). Besides, the greatly extended shelf life was also applicable to strawberry juice (Supplementary Fig. 67) and tomato juice (Supplementary Fig. 68). In addition, compared with the standard sterilization methods in food industry, the proposed TIF-based method showed comparable sterilization performance as pasteurization, but was much better than the O$_3$ treatment (Fig. 7d). It should be noted that the TIF-based method was efficient in treatment of acid- and heat-resistant *A. acidoterrestris* (Fig. 6), the main cause of fruit juice spoilage and deterioration (even after pasteurization[41]). These data confirmed the excellent photodynamic sterilization performances of TIF in acidic fruit juices, which may change the transport mode of commercial fruit juices from production region to sale places, and thus dramatically reduce the costs.

Next, the influence of the added TIF to jucies after photodynamic sterilization was further investigated from the following two aspects. First, like many other dyes, TIF experienced photo-bleaching upon photo irradiation. As shown in Supplementary Fig. 69, in the presence of green LED irradiation (520 nm, 50 mW/cm²), the absorbance of TIF decreased rapidly in 5 min (with ~22% remaining). In other words, most of the added TIF in juices is expected to lose the ability of ROS generation after the first photodynamic sterilization process. Second, the ingredients inside the juices before and

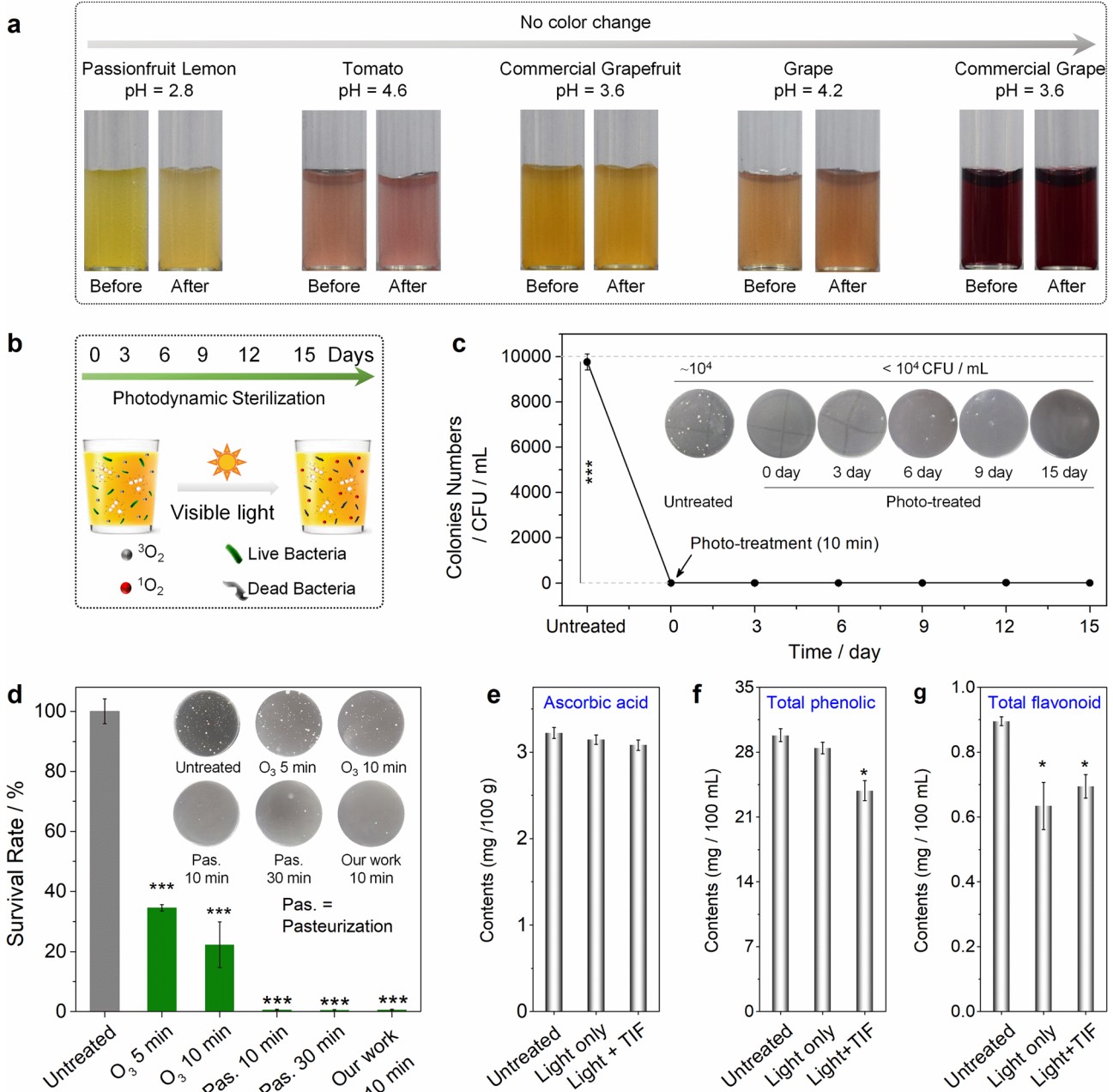

**Fig. 7 TIF for photodynamic sterilization of acidic fruit juices. a** Pictures showing the color change of juices treated with TIF before and after light irradiation; **b** schematic illustration of using TIF for photodynamic sterilization in juice under visible light irradiation; **c** bacterial colony number in passionfruit lemon juice (pH = 2.8) treated with TIF-based photodynamic sterilization at different time intervals, with pictures of the bacterial cultures shown inset (bacteria were first treated and then plated on solid media for colony quantification); **d** comparison of the performances of different sterilization methods for treatment of passionfruit lemon juice; **e** change of the ascorbic acid concentrations in passionfruit lemon juice before and after photodynamic sterilization; **f** change of the total phenolic compounds concentrations in passionfruit lemon juice before and after photodynamic sterilization; and **g** change of the total flavonoid concentrations in passionfruit lemon juice before and after photodynamic sterilization (*$P < 0.05$, **$P < 0.01$, ***$P < 0.001$). Error bars represent standard deviation, $n = 3$ independent replicates.

after TIF-involved photodynamic sterilization were evaluated. Here, the investigations were concentrated on antioxidants due to the high activity of ROS[42]. Three representative antioxidants, namely, ascorbic acid (vitamin C), phenolic compounds, and flavonoid, were chosen. As shown in Fig. 7e–g, the contents of these antioxidants were lowered after photo-treatment (*$P < 0.05$, Supplementary Figs. 70–72). Compared with the common operations in juices production (e.g., thermal treatment, concentrate, and storage)[43–46], the changes of the antioxidants by photodynamic sterilization with TIF were comparable or lower

(Supplementary Table 9). It should be noticed that only 10–20 min of light irradiation (once) was employed for all the above photodynamic sterilization investigations. Besides, the color, flavor, and texture of the juices were roughly evaluated through visual observation and smelling (method supplied in the Chinese National Standard GB/T 31121-2014), no appreciable changes were observed before and after TIF-based photodynamic sterilization. Moreover, the changes of the three antioxidants under real sunlight irradiation were investigated to simulate poential light irradiation during storage and transportation of the

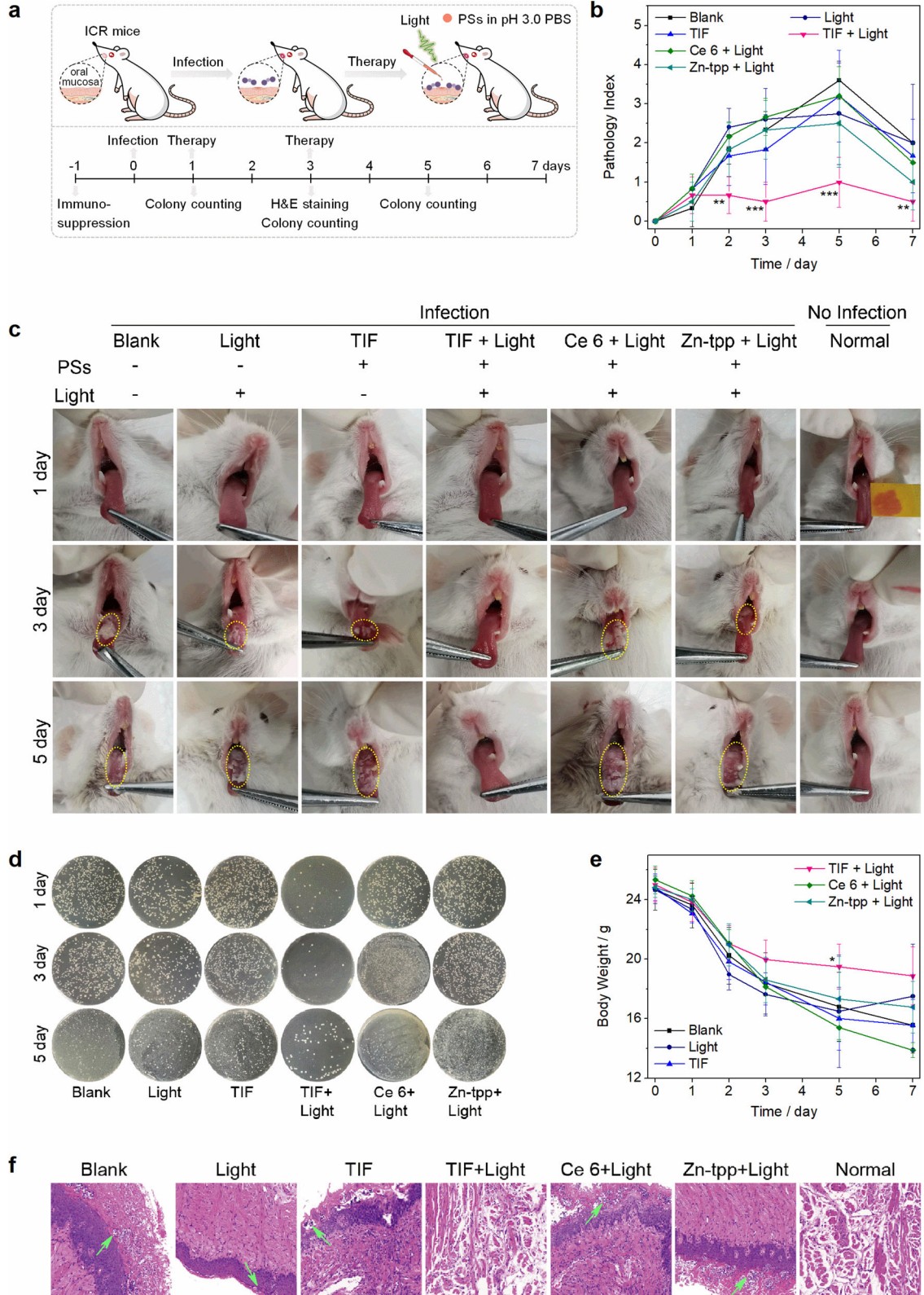

**Fig. 8 TIF for in vivo PACT of oral Candidiasis under extremely acidic condition. a** Schematic illustration of the infection and therapeutic process; **b** pathology index analysis of the mice of different groups at different time intervals (**$P < 0.01$, ***$P < 0.001$). Error bars represent standard deviation, $n = 6$ independent replicates; **c** photographs of the oral mucosal infection on tongue of the mice, the inset image in the normal group revealed pH on the tongue (pH test paper); **d** fungal burdens of different groups (bacteria was separated from oral mucosa and then cultured on agar plates); **e** body weight of mice with/without treatment at different time intervals (*$P < 0.05$). Error bars represent standard deviation, $n = 6$ independent replicates; and **f** photomicrographs showing the section of tongues of mice with H&E staining.

jucies. As can be seen in Supplementary Fig. 73, after 8 h of continuous sunlight exposure (maximum and average irradiance of 25.8 and 7.8 mW/m², respectively), the contents of these nutrients changed only slightly. Therefore, TIF-based photo-dynamic sterilization is appealing for the acidic food preservation.

Besides fruit juices, the potential of TIF for photo-assisted fresh food preservation was also investigated with tomato as the model (pH 3.0). As shown in Supplementary Fig. 74, except the TIF + light group, different magnitudes of wound infection were observed for the groups of blank (no TIF and light), light only, TIF only, Ce 6 + light, and Zn-tpp + light. Meanwhile, the colony counting of the infected tomatoes in 3 day also prove the excellent antimicrobial ability of TIF. Therefore, the food additive TIF is also promising for fresh food preservation.

**In vivo photodynamic antimicrobial chemotherapy of oral Candidiasis**. Candidiasis is an infection caused by *Candida*, a fungal normally lives in places, such as mouth, throat, gut, and vagina. Particularly, due to the widespread use of antibiotics, glucocorticoid, and immune suppressers, as well as the rapid increase of HIV/AIDS infectors, and organ transplant and diabetes patients, the incidence of Candidiasis raised greatly, especially in immunocompromised persons[47,48]. Therefore, Candidiasis therapy is important, but medication of which with antifungal drugs is relatively difficult due to the similarities of fungal cells and mammalian cells. Besides, *C. albicans* could also be colonized in gastric mucosa (pH < 3) and play a synergistic pathogenic role with *H. pylori*[49]. Here, we explored the in vivo potential of TIF for extremely acidic photodynamic antimicrobial chemotherapy (PACT) of Candidiasis.

In order to stimulate the *C. albicans* infection in gastric mucosa, oral mucosa of *ICR* mice was chosen as the infection model for direct and facile visualization of the infection and the photodynamic therapeutic effect of TIF. The extremely acidic environment was constructed through changing the pH of the mucosa and diluting PSs in pH 3.0 PBS buffer. As shown in Fig. 8a, after immune suppression (day −1), the oral mucosa of mice was first infected with *C. albicans* (day 0), and then subjected to photodynamic antimicrobial chemotherapy at day 1. To maximize the phototherapeutic difference between TIF and the control PSs (Ce 6 and Zn-tpp) and also consolidate the therapeutic effect, the infected mice were treated again at day 3. During the investigations, the infected *ICR* mice were divided into six groups (six mice per group): blank, light, TIF, TIF + light, Ce 6 + light, and Zn-tpp + light. For comparison and better visualization of the infection, a normal group (no infection) was also included.

After infection, a white pseudo membrane was occurred on the tongue of the mice (day 1, Fig. 8c and Supplementary Fig. 75), with slightly increased pathology index (Fig. 8b), indicating successful *C. albicans* infection. Upon PACT with TIF (the TIF + light group), the pathology index remained almost unchanged (Fig. 8b) and the pseudo membrane disappeared gradually (Fig. 8c and Supplementary Fig. 75). Whereas for the rest groups, the area and thickness of the pseudo membrane increased largely during day 1 to day 5. Particularly at day 5, except the TIF + light group, the area of the pseudo membrane of other infected groups was even >90% (maximum pathology index over 4, Fig. 8b). The fungal burdens of the corresponding groups were consistent with the results of pathology index and infection images (Fig. 8d). Meanwhile, due to the decreased appetite of mice by oral mucosal infectious, the body weights of the infected mice were also decreased gradually, but the TIF + light group experienced the least loss among the investigated groups within 7 days (*P < 0.05, Fig. 8e). Further pathologic investigations with H&E staining indicated that large amounts of inflammation infiltration could be identified in the groups of blank, light, TIF, Ce 6 + light, and Zn-tpp + light (Fig. 8f), while the case of the TIF + light group was generally similar to that of the normal group (no infections). Overall, these results confirmed the promising PACT performance of TIF to infections under extremely acidic conditions.

## Discussion

In summary, by taking advantage of the pH responsiveness of the fluorescein core structure (spirocyclization) and the HAE of halogens, here we reported an interesting concept for developing PSs that can work in extremely acidic conditions. Upon halogenation, lowest $pH_{0.5}$ of 2.79 was observed (TCF-1), resulting in the efficient absorption in extremely acidic pH range. Meanwhile, maximum singlet oxygen generation efficiency of 0.75 (RB) was obtained. The dual effects of halogenation yielded a series of efficient PSs for extremely acidic photodynamic bacteria inactivation (e.g., PB, RB, and TIF), as demonstrated with *L. plantarum*. Further successful treatment for a broad band of aciduric bacteria, including Gram-positive (*L. plantarum* and *S. aureus*), gram-negative (*E. coli, Salmonella,* and *H. pylori*), and fungus (*C. albicans*), confirmed the excellent performance of TIF ($IC_{50}$ < 1.1 µM). Since TIF is an FDA-approved food additive, its photodynamic sterilization performance was further explored for inhibition of bacteria growth in acidic beverages. Greatly extended shelf life of acidic beverages was observed, showing the appealing feature of TIF photodynamic sterilization for low toxicity food preservation. Moreover, successful in vivo PACT for oral Candidiasis with TIF was achieved, which greatly expand the application of the proposed extremely acidic PSs.

## Methods

**pH titration of halogenated fluorescein**. Fluorescein derivatives (Supplementary Table 1) were dissolved in DMSO to obtain 1 mM stock solutions, and then diluted with citric acid–Na₂HPO₄ buffer (100 mM) to 10 µM. The pH of the solutions were adjusted through varying the ratios of citric acid and Na₂HPO₄ (pH 2.2–8.0), and further acidity was adjusted with concentrated HCl or 1 M NaOH. The absorption and fluorescence spectra of the resultant solutions were then collected.

**Determination of the relative singlet oxygen quantum yields of different fluorescein-based photosensitizers**. The singlet oxygen quantum yields ($\Phi_\Delta$) of all fluorescein-based PSs were evaluated with Rose Bengal as the standard ($\Phi_\Delta$ = 0.75):

$$\Phi_\Delta = \frac{A_s(\text{Absorption})}{A_s(\text{Emission})} \times \frac{A_{RB}(\text{Emission})}{A_{RB}(\text{Absorption})} \times 0.75 \qquad (1)$$

Here, the area of absorption (RB and PSs) was collected with an integrating sphere ($\lambda$ex = 510 nm), while the area of emission was obtained from the characterized 1270 nm $^1O_2$ phosphorescence emission. For collection of such emission, CH₃CN and D₂O mixed solvent ($\nu$ (D₂O): $\nu$ (CH₃CN) = 1:15, pH = 9.0, adjusted with 0.001 M NaOH) was used, since H₂O can severely quench the singlet oxygen luminescence. For all the measurements, the concentrations of the PSs were set at 10 µM.

**$^1O_2$ phosphorescence emission under different pH and irradiance**. A total of 10 µM TIF solutions were first prepared with citric acid–Na₂HPO₄ buffer (100 mM, diluted by D₂O) to yield different pH (2.2–6.0). Then, a laser of 532 nm was selected as the lighting source, and the $^1O_2$ phosphorescence emission were collected.

**Culture and inhibition of *L. plantarum* with TIF**. The *L. plantarum* cells were anaerobically cultured in 5.24% MRS broth in a constant temperature oscillator (37 °C, 170 r/min) for 12 h. Afterward, the *L. plantarum* bacterial liquid was diluted one million times with PBS buffer (pH 7.4). For inhibition at different pH, the activated bacterial liquid was diluted by pH 7.0, 4.0, 2.5 PBS buffer, respectively. Then, the PSs (0.5 µM) were added and the solution were irradiated with LED (520 nm, 3 V, 3 W). Finally, the treated bacterial liquid (100 µL) was smeared onto agar plants (containing 5.24% MRS broth and 1.5% agar) and cultured at 37 °C for another 48 h. The colony numbers of *L. plantarum* were counted, and the survival rate were calculated as $C/C_0 \times 100\%$, where C is the CFU of sample treated with PSs and light, and $C_0$ is the blank without any treatment.

**Evaluation of IC$_{50}$ of different aciduric bacteria**. Bacterial strains are listed in Supplementary Table 6. *S. aureus* and MRSA, and *C. albicans* were provided by West China Hospital of Stomatology, Sichuan University. *A. acidoterrestris*, *S. enterica*, *H. pylori*, *L. plantarum*, and *E. coli* were provided by College of Life Science, Sichuan University. The bacteria were cultured overnight in a constant temperature oscillator (37 °C, 170 r/min, the specific culture conditions were listed in Supplementary Table 6). Suspensions of bacteria (100 μL, diluted 1000-folds, PBS) were added with different concentration PSs (final volume of 1 mL), and irradiated with LED ($\lambda = 520$ nm, 35 mW/cm$^2$) for 10 min. Then, the treated bacteria liquid (100 μL) were added into the culture medium and cultured in a 24-well plate. The OD$_{600}$ of bacteria was measured to obtain the bacteria CFUs, and the survival rate were calculated as $C/C_0 \times 100\%$, where $C$ is the CFU of sample which treated with PSs and light, and $C_0$ is the blank without any treatment. All photodynamic sterilizations were carried out in at least three replicates.

**Photodynamic sterilization of acidic juices**. Fresh juices were prepared and divided into two groups (each to a culture dish of 60 mm id), and then TIF was added with final concentration of 10 μM. One group was subjected to light processing (xenon lamp, 300 W, 420 nm optical filter, beam spot size 60 mm) for 10–20 min, and another without. All samples were sub-packed into the headspace bottle and stored in 4 °C for different time. To determine the bacteria colonies, 1 mL juice samples were mixed with plate count agar and culture at 37 °C for 48 h (GB 4789.2-2016). Afterward, the total number of bacteria colonies were counted.

**Fresh food preservation**. Tomatoes were selected as the model for photo-assisted fresh food preservation[50]. All the tomatoes were divided into six groups, and made a wound of ~7 cm$^2$ by hot water. Then, gauzes immersed in solutions containing 10$^8$ CFU/mL *C. albicans* were placed on the wound sites. After infection for 1 h, 300 μL pH 3.0 PBS or PSs (10 μM, diluted by pH 3.0 PBS) were added to wounds, and further irradiated with LED (520 nm for TIF, 620 nm for Zn-tpp, and 660 nm for Ce 6; irradiance: 35 mW/cm$^2$) for 10 min. Afterward, the tomatoes were stored at room temperature and then subjected to repeated irradiation at day 2. The total viable counts of bacteria on the infected tomatoes were measured in day 3.

**Photodynamic antimicrobial chemotherapy of TIF in vivo**. Healthy female *ICR* mice (20–25 g) were chosen in this work. All of the performed experiments were approved by the Subcommittee on Research and Animal Care of Sichuan University. The mice were reared in separate cages (six mice per cage), and adapted for 1 week free to drink and eat at room temperature. To decrease the self-limitation of mice toward *C. albicans*, immune-suppressor prednisolone (125 mg/kg) were intramuscular injected at day −1 and day 3 relative to the infection. In order to prevent unknown bacterial infection, 0.83 g/L tetracycline hydrochloride was added to drinking water for mice. To construct the oral mucosal infection model, cotton balls immersed in solutions containing 10$^8$ CFU/mL of *C. albicans* were placed in the mouth of mice for 75 min. At day 1 and day 3, 50 μL pH 3.0 PBS and PSs (20 μM, diluted by pH 3.0 PBS) were added to oral mucosa, and treated with different lasers (510 nm for TIF, 660 nm for Ce 6 and Zn-tpp, 50 mW/cm$^2$) for 10 min.

To investigate the therapy effect, photographs, colonies, weights, and pathology index of mice were taken every other day. After day 3, one mouse in each group was sacrificed, and the tissues of oral mucosa were harvested for hematoxylin and eosin (H&E) staining. The pathology index was evaluated based on the standard given in Supplementary Table 10.

**Statistical analysis**. Error bars represent the standard deviation derived from three independent measurements, and all the statistical analysis was performed using Microsoft Excel. The images were processed with Microsoft PowerPoint and Origin software. The statistical significance of differences was determined by a One Way ANOVA, and *$P < 0.05$, **$P < 0.01$, and ***$P < 0.001$ were used to indicate the statistical difference.

**Reporting summary**. Further information on research design is available in the Nature Research Reporting Summary linked to this article.

## Data availability

The authors declare that the data supporting the findings of this study are available within the paper and the Supplementary Information. All data are available from the authors on reasonable request.

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

## Acknowledgements

We gratefully acknowledge the financial support from the National Natural Science Foundation of China (Nos. 21522505 and U19A2005), Multi-Disciplinary Research Program of Sichuan University (No. 2020KXK0404), and the Fundamental Research Funds for Central Universities (No. 2018SCUH0075). Detailed characterizations supported by the public Platform of Analytical and Testing Center, Sichuan University, are greatly appreciated.

## Author contributions

P.W. supervised this work. Y.W. and J.L. prepared and characterized the photo-sensitizers, and carried out all the photodynamic sterilization experiments. Y.W., J.L., and R.Z. performed the PACT investigations. Z.Z. and Q.S. helped culturing the bacteria. Y.W. and P.W. wrote the manuscript.

## Competing interests

Y.W. and P.W. are inventors on the pending Chinese patent application 202010141047.5, filed by Sichuan University related to this work on 4 March 2020. Other authors declare no competing interests.

## Additional information

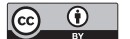

