## [Peer Review File · Nature Communications]

Reviewers' comments:

Reviewer #1 (Remarks to the Author):

This manuscript demonstrates the use of halogenated fluorescein derivatives for photodynamic antibacterial effect in acidic pH. The photosensitizer showed great growth inhibition especially for acid-fast bacteria, suggesting a potential of the dye as an alternative for food sterilization. The overall concept of this study is pretty interesting; however, the feasible use of this photosensitizer is not convincing, and there are some contradictions for this compound as a food additive. Moreover, this study would have better fit for the publication in more specified journals rather than Nature Communications. Therefore, this study should be submitted to more specialized journals such as scientific report.

1. Introduction part describes the needs of a photosensitizer that works in acidic environment because of the acidic nature of our stomach. However, overall applications of this study are mostly nothing to do with the acidity of the stomach, but just the acidity of the food itself (e.g. fruit juice).
2. For the use as a food additive, the solubility of the photosensitizer seems problematic. As described in methods section, the dye requires CH₃CN (acetonitrile), a toxic solvent and carcinogen, in order to solubilize it as a food additive.
3. Generation of reactive oxygen species will oxidize several ingredients in the food. This might affect not only the taste of the food but the active compounds through the oxidization process. Do authors have any remark on this?
4. Due to the aforementioned two major concerns (comment #2 and #3), one promising alternative for the application of this photosensitizer is to apply it for actual antibacterial therapy after the oral administration (this would be very interesting). In that case, two points have to be clearly addressed; 1) the light source should be the one with longer wavelength, 2) the presentation of convincing in vivo data.

Reviewer #2 (Remarks to the Author):

Wang et al. demonstrated that a novel photosensitizer, namely halogenated fluorescein, inhibited bacteria growth of a broad band of acid-fast bacteria upon illumination with visible light and that this approach suppressed bacteria growth in an acidic condition. The approach is novel; chemistry is solid; their conclusions are well supported with the ample sets of data.

The manuscript can be further improved by providing technical information on the light exposure and statistical method used.

The manuscript hints the potential of the halogenated fluorescein for photodynamic anti-microbial chemotherapy (PACT), but the study mainly focused on disinfection of food products. Although the reviewer agrees to the idea that this technology could be further explored for disinfection of blood products and treatment of infection, this point has to be clarified.

1. The authors claim that a halogenated fluorescein could be useful for PACT. However, no relevant in vivo experiment was performed in the context of PACT. Along with the claim, the authors should provide experimental evidence that this could be used for PACT in a relevant model such as localized infection in animals, preferably at acidic parts of the body. If the main message of this work is to create a new sterilization method for food packaging, the effect of this approach needs to be compared to the standard method using ionizing radiation such as gamma rays, x-rays or electron beams.
2. The authors used visible (in Figure 3-4) or visible-near-infrared (in Figure 5) light for this approach, which has limited and variable penetration depth into samples depending on what kinds

of chromophores food products contain. In Figure 3-4, the authors chose green 520 nm light, while they used light from Xenon lamp filtered with a 420 nm long-pass filter. Please provide the reason for these choices and discuss the impact of this difference in the light source on the anti-microbial effect.

3. Since the effect of this approach could be dependent on light penetration depth, the detailed technical description on the light illumination is necessary. In Figure 4, 520 nm light was used at an irradiance of 6 mW/cm², while light from Xenon lamp filtered with a 420 nm long-pass filter was used at an irradiance of 100 mW/cm² for Figure 5. In order to understand the light penetration depth into samples in these particular settings, please provide the detailed pieces of information including the dimension of container of the samples, beam spot size on the samples, distribution of power across the illumination field.

Minor

- 1) "Acid-fast" is used inconsistently between "acid-fast" and "acid fast" throughout the manuscript.
- 2) In page 2 in the abstract, the authors state "...far lower than the maximum level in food guided by Chinese National Standard (57 μM)". Since TIF is also approved by FDA, it is helpful to include a statement comparing it with the FDA standard level as well.
- 3) Please provide which parameter of light (wavelength, irradiance) used for the Figure 3 experiment.
- 4) Across the Figures, there is no description how many times these particular experiments were performed to confirm reproducibility.
- 5) In Figure 3, 5: the authors describe no statement on which statistical method was used to compare differences across groups. Please clarify how the authors perform the statistical analysis for each study. In some experiments, there are more than 2 samples crossed with factors; please clarify which method was used to correct for multiple comparisons across groups.

Response to Reviewers' Comments

Dear reviewers:

We highly appreciate the reviewers for your constructive comments. Although we placed our detailed response to comments and suggestions below, it is difficult to indicate the place where reversions made, since the MS was thoroughly revised and almost re-written. By the way, **all changes have been red-marked in the revised manuscript.**

Reviewer #1

Remarks to the Author:

This manuscript demonstrates the use of halogenated fluorescein derivatives for photodynamic antibacterial effect in acidic pH. The photosensitizer showed great growth inhibition especially for acid-fast bacteria, suggesting a potential of the dye as an alternative for food sterilization. The overall concept of this study is pretty interesting; however, the feasible use of this photosensitizer is not convincing, and there are some contradictions for this compound as a food additive. Moreover, this study would have better fit for the publication in more specified journals rather than Nature Communications. Therefore, this study should be submitted to more specialized journals such as scientific report.

Reply: Thanks very much for your positive comments!

The MS was thoroughly revised according to the suggestions from two reviewers. The essences for the current reversion were as follows:

1. We intended to develop photosensitizers that can work at extremely acidic conditions. Eventually, among the discovered photosensitizes (PB, TIF, and RB), we unexpectedly found that TIF was an approved food additive. Therefore, we explored its use of photodynamic sterilization in acidic juices, the pH of which is in the extremely acidic range (pH < 4.0).
2. As detailed in our MS, the photodynamic sterilization performance of TIF in acidic juices was satisfactory, which may eliminate the illegal use of preservatives and is

- also economically appealing (patent appealing). Considering the importance and widespread attention of food safety, the application explored in this MS is appealing.
3. Previously, the term “photodynamic antimicrobial therapy (PACT)” was used in the title and throughout the MS. We noticed that “therapy” may be inappropriate and misleading, since no related work was carried out. Besides, the absorption of the photosensitizers discovered here is lower than 600 nm (majorly visible range), which is not suitable for therapy applications due to the issue of light penetration depth. We thus used “photodynamic bacteria inactivation” or “photodynamic sterilization” throughout the MS.

Therefore, we still focused on photodynamic sterilization applications in acidic juices.

1. Introduction part describes the needs of a photosensitizer that works in acidic environment because of the acidic nature of our stomach. However, overall applications of this study are mostly nothing to do with the acidity of the stomach, but just the acidity of the food itself (e.g. fruit juice).

Reply: Thanks very much for your valuable comments!

Here, we referred stomach since we wanted to highlight the difference between “extremely acidic” and “normally acidic”, which can help strengthening the importance of this work. We agreed that “stomach” here may be misleading, since no related applications were carried out. The contents about “stomach” were thus deleted. Besides, as indicated above, the title and the MS was revised to eliminate the use of “therapy”.

2. For the use as a food additive, the solubility of the photosensitizer seems problematic. As described in methods section, the dye requires CH₃CN (acetonitrile), a toxic solvent and carcinogen, in order to solubilize it as a food additive.

Reply: All the photosensitizers tested in this MS is highly water-soluble. Here, acetonitrile was used only as a testing media for better collection of the characterized singlet oxygen phosphorescence emission (1275 nm), since H₂O can severely quench the singlet oxygen luminescence (for details, see the reference below).

To avoid potential misleading, the related parts was revised as follows (Page S31-32, SD):

“For better collection of the characterized singlet oxygen phosphorescence emission (1275 nm), CH₃CN and D₂O mixed solvent was used (H₂O can severely quench the singlet oxygen luminescence)”

When evaluating the pH-dependence of the ¹O₂ generation, we only used D₂O to dilute the sample and a 532 nm laser was thus used as the excitation source for boosting the 1275 nm emission. The related contents was revised as follows (Lines 12-14 in Page 22):

“10 μM TIF solutions were first prepared with citric acid-Na₂HPO₄ buffer (100 mM, diluted by D₂O) to yield different pH (2.2-6.0).”

Ref.: Ogilby, P. R.; Foote, C. S., Chemistry of singlet oxygen. 42. Effect of solvent, solvent isotopic substitution, and temperature on the lifetime of singlet molecular oxygen (1.DELTA.g). *J. Am. Chem. Soc.* **1983**, *105* (11), 3423-3430.

While for all the photodynamic inactivation investigations in this work, the photosensitizers were solubilized in the corresponding buffer. For example, the solubility of TIF can be visualized from Figure R1 below.

Figure R1. The color pictures of the aqueous solutions containing different TIF concentrations (water media).

3. Generation of reactive oxygen species will oxidize several ingredients in the food. This might affect not only the taste of the food but the active compounds through the oxidization process. Do authors have any remark on this?

Reply: Thanks very much for your comments!

Since ROS generated from the photodynamic process are highly active, we thus first evaluated the changes of several representative antioxidants, including ascorbic acid (vitamin C), phenolic compounds, and flavonoid. As can be seen from Figure R2 below, the contents of these antioxidants were lowered after photo-treatment, but was insignificant. It should be noticed that only 10-20 min of light irradiation (once) was employed for all the above photodynamic sterilization investigations.

Figure R2. The changes of the antioxidants before and after TIF-based photodynamic sterilization in passionfruit lemon juice (pH = 2.8).

Besides, the status of the juices (color, fragrance, and texture) were evaluated according to the methods supplied in the Chinese National Standard (GB/T 31121-2014, Fruit & vegetable juices and fruit & vegetable beverage). Currently, we didn't found appreciable change of the juices before and after TIF-based photodynamic sterilization.

The above information was added in the Section of “Photodynamic sterilization of Acidic Juices with TIF”. Besides, Figure 5 was also revised with added information of ascorbic acid, phenols, and flavonoid concentrations.

4. Due to the aforementioned two major concerns (comment #2 and #3), one promising alternative for the application of this photosensitizer is to apply it for actual antibacterial therapy after the oral administration (this would be very interesting). In that case, two points have to be clearly addressed; 1) the light source should be the one with longer wavelength, 2) the presentation of convincing in vivo data.

Reply: Thanks very much for your suggestions!

As stated above, your comment #2 and #3 have already been addressed.

We agreed that photosensitizers for actual antibacterial therapy after the oral administration would be interesting. However, just as the reviewer stated, a prerequisite is the lighting source, which should be long enough to afford the penetration depth. Currently, the absorption of the photosensitizers discovered in this work is lower than 600 nm (majorly visible range). Therefore, we think the photosensitizers in this work may be not suitable for therapy-based applications.

Since one of the discovered photosensitizer, TIF, is an FDA-approved food additive, we thus focused on photodynamic sterilization applications in acidic juices. It's well-known that the fruit quality grow in near-desert areas (e.g., Turpan, China) is much higher than other areas. In order to preserve the freshness, air shipping is exclusively used to deliver the fruits to the destination (e.g., Beijing, China, Figure R3 below), which will largely increase the whole costs. As stated in our MS, the shelf life of the acidic juices could be greatly extended (~15 days), which may allow shipping with motor or rail (to almost all cities in China) and dramatically reduce the costs. A patent application of the related usage was already pending.

Figure R3. Characteristics of different transportation form production regions to destination.

Reviewer #2

Remarks to the Author:

Wang et al. demonstrated that a novel photosensitizer, namely halogenated fluorescein, inhibited bacteria growth of a broad band of acid-fast bacteria upon illumination with visible light and that this approach suppressed bacteria growth in an acidic condition. The approach is novel; chemistry is solid; their conclusions are well supported with the ample sets of data.

The manuscript can be further improved by providing technical information on the light exposure and statistical method used.

The manuscript hints the potential of the halogenated fluorescein for photodynamic anti-microbial chemotherapy (PACT), but the study mainly focused on disinfection of food products. Although the reviewer agrees to the idea that this technology could be further explored for disinfection of blood products and treatment of infection, this point has to be clarified.

Reply: Thanks very much for the positive comments!

Here, the photosensitizers discovered here is lower than 600 nm (majorly visible range), which is not suitable for therapy applications (e.g., disinfection of blood products and

treatment of infection). Therefore, we revised the MS and deleted “photodynamic antimicrobial therapy (PACT)”. Since one of the discovered photosensitizer, TIF, is an FDA-approved food additive, we thus still focused on photodynamic sterilization applications in acidic juices.

1. The authors claim that a halogenated fluorescein could be useful for PACT. However, no relevant in vivo experiment was performed in the context of PACT. Along with the claim, the authors should provide experimental evidence that this could be used for PACT in a relevant model such as localized infection in animals, preferably at acidic parts of the body. If the main message of this work is to create a new sterilization method for food packaging, the effect of this approach needs to be compared to the standard method using ionizing radiation such as gamma rays, x-rays or electron beams.

Reply: Thanks very much for your valuable comments!

As indicated above, animal-related applications were not carried out due to the inefficient absorption of the discovered photosensitizers. We thus focused on the development of a new sterilization method for acidic food preservation.

Currently, the most often used method for food sterilization is thermal sterilization (e.g., pasteurization). Other methods, such as radiation sterilization and O₃ treatment, are also used. Generally, ionizing radiation is majorly used for solid foods. For safety considerations, here we compared the performance of our method with pasteurization and O₃ treatment. As shown in Figure R4 below (Figure 5D), the proposed TIF method showed comparable sterilization performance as pasteurization, but was much better than the O₃ treatment. It should be noted that the TIF-based method was efficient in treatment of acid- and heat-resistant *Alicyclobacillus* (see Figure 4, IC₅₀ of 0.58 μM), the main cause of fruit juice spoilage and deterioration.

Ref.: Smit, Y., Cameron, M., Venter, P. & Witthuhn, R. C. *Alicyclobacillus* spoilage and isolation – A review. *Food Microbiol.* **28**, 331-349.

Figure R4. Comparison of the sterilization performances of TIF-based photodynamic inactivation with pasteurization and O₃ treatment.

The above information was added in the Sections of “Photodynamic inactivation with TIF for aciduric bacteria” and “Photodynamic sterilization of Acidic Juices with TIF”. Besides, Figure 4 and Figure 5 was also revised with added information of Alicyclobacillus and method comparisons.

2. The authors used visible (in Figure 3-4) or visible-near-infrared (in Figure 5) light for this approach, which has limited and variable penetration depth into samples depending on what kinds of chromophores food products contain. In Figure 3-4, the authors chose green 520 nm light, while they used light from Xenon lamp filtered with a 420 nm long-pass filter. Please provide the reason for these choices and discuss the impact of this difference in the light source on the anti-microbial effect.

Reply: Thanks very much for valuable suggestion!

In Figure 3 and Figure 4, the performances of TIF-based photodynamic inactivation were evaluated for aciduric bacteria, thus model bacteria were used here (Lactobacillus for Figure 3 and a broad band of bacteria for Figure 4). All the bacteria were cultured in plate and we thus used 520 nm green LED for ease of operation, which matched the absorption maxima of TIF.

For Figure 5, near real-world applications (fruit juices) were carried out, we thus tested white light for compatibility. As shown in Figure R5 below (Figure S68), there is no significant difference of antimicrobial performances between white and green LED irradiation, indicating that white light can still be used. The choice of Xenon lamp filtered with a 420 nm long-pass filter here was mainly because it was most often used in the literature as the white light sources.

Currently, we didn't observe appreciable influence of the light penetration depth, probably because the juice samples investigated here were almost transparent. Of course, the power density of the Xenon lamp may also influence the exact irradiance of a specific penetration depth. For future industrial applications, the conditions of the light sources should be optimized based on the real scenes, but the use of white light can be confirmed here.

To illustrate such difference, the detailed information about the Xenon lamp was added in the text as follows (lines 15-18 in Page 17):

“Here, a Xenon lamp equipped with a 420 nm long-pass filter was chosen as the light source to simulate sunlight for subsequent studies, the power density (100 mW/cm^2) of which may also permit higher irradiance over green LED at a specific penetration depth.”

Figure R5. Survival rates of microbial in passion fruit lemon juice after treated with different light source (520 nm green and white LED, 23 mW/cm^2 , * $P < 0.05$, ** $P < 0.01$, *** $P < 0.001$).

3. Since the effect of this approach could be dependent on light penetration depth, the detailed technical description on the light illumination is necessary. In Figure 4, 520 nm light was used at an irradiance of 6 mW/cm², while light from Xenon lamp filtered with a 420 nm long-pass filter was used at an irradiance of 100 mW/cm² for Figure 5. In order to understand the light penetration depth into samples in these particular settings, please provide the detailed pieces of information including the dimension of container of the samples, beam spot size on the samples, distribution of power across the illumination field.

Reply: Thanks very much for your comments!

The dimension of container of the samples (culture dish, 60 mm id) and the beam spot size (60 mm) were added in the experimental section (page 23, Photodynamic sterilization of Acidic Juices with TIF).

Since the beam spot size matched with the dimension of the sample container, we think the distribution of the power across the illumination field may be uniform.

Minor

1) “Acid-fast” is used inconsistently between “acid-fast” and “acid fast” throughout the manuscript.

Reply: Thanks! After searching the literature, we used “aciduric bacteria” throughout the MS.

2) In page 2 in the abstract, the authors state “...far lower than the maximum level in food guided by Chinese National Standard (57 μM)”. Since TIF is also approved by FDA, it is helpful to include a statement comparing it with the FDA standard level as well.

Reply: Thanks! The FDA standard limit of TIF was given as Good Manufacturing Practice (GMP), so we chose Chinese National Standard as a standard (Table S6).

Table S6. The information about the food additive TIF in the national standards of

various agencies.

Agencies	Name	Maximum use level	Standards
US FDA	FDC Red No 3	GMP	CFR-Title 21, Part 74
European Union	Erythrosine BS	150 mg/kg	European Parliament and Council Directive 94/36/EC ANNEX III
Chinese National Standard	Erythrosine B	50 mg/kg	GB 2760-2011

3) Please provide which parameter of light (wavelength, irradiance) used for the Figure 3 experiment.

Reply: Thanks!

For TIF-based photodynamic sterilization, 520 nm green LED (3V, 3W) was used. While for Zn-tpg and PC, 620 nm red LED was used. Such information was added in the caption of Figure 3 (Page 12).

4) Across the Figures, there is no description how many times these particular experiments were performed to confirm reproducibility.

Reply: Thanks!

In this work, all of the photodynamic sterilization investigations were carried out with at least three replicates (Figure 3, Figure 4, Figure 5, and related contents in Supporting Information). Besides, error bars were added in Figure 4.

5) In Figure 3, 5: the authors describe no statement on which statistical method was used to compare differences across groups. Please clarify how the authors perform the statistical analysis for each study. In some experiments, there are more than 2 samples crossed with factors; please clarify which method was used to correct for multiple comparisons across groups.

Reply: Thanks very much for your comments!

The differences of the data in Figure 3 and Figure 5 were calculated with significance tests, and the related information was added (*P < 0.05, **P < 0.01, ***P < 0.001) in the captions for Figure 3 and Figure 5.

Reviewers' comments:

Reviewer #1 (Remarks to the Author):

This manuscript, a revised version of previous one, showcases the use of halogenated fluorescein derivatives for photodynamic sterilization in acidic juices. Although the authors have tried to respond to all technical questions that commented by reviewers and thus significant changes have been made throughout the revised manuscript, the impact of the application model of TIF is still too specified and limited for being consideration for Nature Communications. In addition, the authors had provided many fancy/polished terms in the title and introduction that potentially could draw interests of readers by misleading the original manuscript, and simply removed all these terms in the revised manuscript, replying that these were not what they meant. As the reviewer mentioned in the previous round, this manuscript should have better fit for Nature Communications only if the compounds reported here were capable of demonstrating antimicrobial chemotherapeutic applications, not as a food additive; otherwise would better to be published in more specialized journals like food-related journal. Therefore, I do not recommend this revised version of manuscript to be considered in this journal any more. Please see below additional comments that might improve the quality of the manuscript further:

1. Since the compound remains in fruit juices as an additive, the reviewer is wondering how long the compound is going to active upon light irradiation. Given the fact that TIF could be activated by sunlight as well, is there any potential concern of ROS generation when those juices are stored outside under sunlight? In other words, having juices exposed under sunlight is going to be safe enough? Toxicity evaluation of the compound toward mammalian cell line is shown only at dark condition (Figure S66).

2. In point 3 of the rebuttal letter, the authors mentioned that the contents of active compounds in fruit juices were lowered but insignificant. It is obvious that the changes were not only lowered but also significant (*) in total phenolic and flavonoid contents according to the statistical analysis run by THE AUTHORS. How did the authors end up getting the conclusion of insignificant changes? Please clarify this.

Reviewer #2 (Remarks to the Author):

The authors substantially revised the paper and addressed all the concerns raised. The manuscript is acceptable.

Although we cannot offer to publish your paper in Nature Communications, the work may be

Responses to the Reviewers' Comments

Manuscript ID: NCOMMS-19-38798A-Z

Title: Halo-Fluorescein for Extremely Acidic Photodynamic Bacteria Inactivation

The comments and suggestions made by the two reviewers are very helpful for us to revise the manuscript. We highly appreciate the reviewers for such constructive comments. Detailed response to the comments and suggestions are made below. **All changes have been red-marked in the revised version of the manuscript.**

Reviewer #1 (Remarks to the Author):

This manuscript, a revised version of previous one, showcases the use of halogenated fluorescein derivatives for photodynamic sterilization in acidic juices. Although the authors have tried to respond to all technical questions that commented by reviewers and thus significant changes have been made throughout the revised manuscript, the impact of the application model of TIF is still too specified and limited for being consideration for Nature Communications. In addition, the authors had provided many fancy/polished terms in the title and introduction that potentially could draw interests of readers by misleading the original manuscript, and simply removed all these terms in the revised manuscript, replying that these were not what they meant. As the reviewer mentioned in the previous round, this manuscript should have better fit for Nature Communications only if the compounds reported here were capable of demonstrating antimicrobial chemotherapeutic applications, not as a food additive; otherwise would better to be published in more specialized journals like food-related journal. Therefore, I do not recommend this revised version of manuscript to be considered in this journal any more. Please see below additional comments that might improve the quality of the manuscript further:

Reply: Thanks very much for the suggestions!

First, for the “fancy/polished terms”, we guess the reviewer mainly criticize on the use of “extremely acidic conditions” and “extreme acidity”. It should be noted that the extreme acidity was not a newly defined phrase in this MS, but in fact already existed in the literature (e.g., Yang et al., Biocompatible click chemistry enabled compartment-specific pH measurement inside *E. coli*, *Nat Commun*, 2014, 5, 4981; Yang et al., Converting a solvatochromic fluorophore into a protein-based pH indicator for extreme acidity, *Angew Chem Int Ed*, 2012, 51, 7674-7679.).

Second, we agree with the reviewer that antimicrobial chemotherapeutic applications are important for potential broad interest of this work. In the first round of reversion, the related investigations were not carried out, partially because of the COVID-19 pandemic, i.e., we had no access to the animal center of our university (also the nearby centers) at that time (March to June).

During the current reversion, we are informed that we can carry out related investigations, as the situation was changed. To demonstrate the antimicrobial chemotherapeutic performance of the extremely acidic photosensitizers, we chose Candidiasis as the model. Candidiasis is an infection caused by *Candida*, a fungal normally lives in places such as mouth, throat, gut, and vagina. Besides, *Candida* could also be colonized in gastric mucosa ($\text{pH} < 3$) and play a synergistic pathogenic role with *H. pylori*. In order to stimulate the *Candida* infection in gastric mucosa, oral mucosa of ICR mice was chosen as the infection model for direct and facile visualization of the infection and the photodynamic therapeutic effect of TIF. During PACT, the pH of the oral mucosa was adjusted to 3 during our investigations. Other infection places (throat, gut, and vagina) were not taken into consider due to the practical difficulty in observation of the infections and the un-suitability of the absorption of the photosensitizers in this work.

As can be seen from Figure R1 below (and also Figure 6 in the revised MS), the photosensitizer TIF exhibited the best antimicrobial performance for mice bearing Candidiasis ($\text{pH} = 3$). Meanwhile, control investigations with commonly used photosensitizers Ce6 and Zn-tpa could hardly work in the extremely acidic media.

Therefore, the in-vivo performance of the photosensitizer TIF was also excellent.

Figure R1. TIF for in vivo PACT of oral Candidiasis under extremely acidic condition: (A) schematic illustration of the infection and therapeutic process; (B) pathology index analysis of the mice of different groups at different time intervals (**P < 0.01, ***P < 0.001); (C)

photographs of the oral mucosal infection on tongue of the mice, the inset image in the Normal group revealed pH on the tongue (pH test paper); (D) fungal burdens of different groups (bacteria was separated from oral mucosa and then cultured on agar plates); (E) body weight of mice with/without treatment at different time intervals (*P < 0.05); and (F) photomicrographs showing the section of tongues of mice with H&E staining.

Besides, since TIF is a food additive, we insisted that the antimicrobial therapeutic performance of TIF for potential fruit preservation was also appealing. Therefore, the data for fruit juices were retained. Moreover, we also carried out investigations with fresh food (with tomatoes as the model). As can be seen from Figure R2 below (and also Figure S67 in the revised MS), TIF exhibited better performances over Ce6 and Zn-tp. Therefore, the developed extremely acidic photo antimicrobial chemotherapy can be also explored for fresh food preservation.

Figure R2. Fruit preservation model of TIF. (A) Photographs of tomatoes with different treatments during 1-3 days. (B) Colony counting of infected tomatoes in 3 day.

We have carefully revised the manuscript, with the above added data. We do hope the revised manuscript is acceptable in its current form.

Comment 1. Since the compound remains in fruit juices as an additive, the reviewer is wondering how long the compound is going to active upon light irradiation. Given the fact that TIF could be activated by sunlight as well, is there any potential concern of ROS generation when those juices are stored outside under sunlight? In other words, having juices exposed under sunlight is going to be safe enough? Toxicity evaluation of the compound toward mammalian cell line is shown only at dark condition (Figure S66).

Reply: Thanks for your comments!

We agree with the reviewer that although the photo-generated ROSs from TIF are beneficial for bacteria killing, their potential influence of TIF itself and the generated ROS to the fruit juices also should be considered. To address this issue, we performed the following investigations:

(1) The photo-bleaching of TIF. Like most dyes, TIF is prone to be photo-bleached. As can be seen from Figure R3 below, in the presence of green LED irradiation (520 nm, 50 mW/cm²), the absorbance of TIF decreased rapidly in 5 min (with ~22% remaining). In other words, most of the added TIF in juices is expected to lose the ability of ROS generation after the first photodynamic sterilization process.

Figure R3. Photo-bleaching test of TIF in pH 3.0 PBS (green LED, 50 mW/cm²).

(2) The change of the contents of the nutrients in juices after sunlight irradiation. Next, the contents of antioxidants (ascorbic acid, total phenolic and total flavone) were

measured to evaluate the loss of nutrient by ROS when juices stored outside under sunlight (8 h, 10:30-18:30, Aug 14, Chengdu, China). As shown in Figure R4, the contents of these nutrients changed only slightly. Considering that the exact concentration of TIF after photodynamic sterilization was lowered, the generated ROS by sunlight would cause no significant effect to the quality of fruit juice after leaving the factory. Besides, it should be noted that commercial juices are mostly stored in the shade, the irradiance of 8 h sunlight would be representative to evaluate the potential concern of ROS generated when those juices are stored outside under sunlight.

Figure R4. Evaluation of the potential influence of sunlight on the contents of the nutrients in passionfruit lemon juice: (A) the changes of irradiance and temperature of the day that the investigations carried out (10:30-18:30, Aug 14, Chengdu, China); (B) the contents of ascorbic acid under sunlight for 8 hours; (C) the contents of total flavonoid under sunlight for 8 hours; and (D) the contents of total phenolic under sunlight for 8 hours.

Therefore, based on the above data, we think the juices exposed under sunlight are safe enough. It should be noted that the conditions taken in this work (Xenon lamp with 420 nm long-pass filter) are somewhat extreme cases, which led to decrease of the nutrients after irradiation. Moreover, considering the uncertainty of sunlight and other conditions, the proposed method is recommended to be carried out in factory under the normal preservation conditions of fruit (normally in a cool and dark place).

The above data and related discussions were added in the revised MS.

Comment 2. In point 3 of the rebuttal letter, the authors mentioned that the contents of active compounds in fruit juices were lowered but insignificant. It is obvious that the changes were not only lowered but also significant (*) in total phenolic and flavonoid contents according to the statistical analysis run by THE AUTHORS. How did the authors end up getting the conclusion of insignificant changes? Please clarify this.

Reply: Thanks for your comment! We apologize that we made a mistake here and misled the reviewer.

In fact, we wanted to indicate here that compared with other operations (such as thermal treatment, concentrate, and storage), the changes of the antioxidants by photodynamic sterilization with TIF were not significant.

The related part was revised as follows:

“As shown in Figure 5E-Figure 5G, the contents of these antioxidants were lowered after photo-treatment (* $p < 0.05$, Figure S77-S79). However, compared with other operations (e.g., thermal treatment, concentrate, and storage), the changes of the antioxidants by photodynamic sterilization with TIF were comparable or lower”.

Reviewer #2 (Remarks to the Author):

The authors substantially revised the paper and addressed all the concerns raised. The manuscript is acceptable.

Reply: Thanks very much for your positive comment!

REVIEWERS' COMMENTS

Reviewer #1 (Remarks to the Author):

In this revised manuscript, authors added another figure related to animal experiment (Fig 6) and responded to the reviewer's comment. Therefore, I think that the authors have made significant improvements to their manuscript and have satisfactorily addressed my concerns. The manuscript is acceptable for publication.

Responses to the Reviewers' Comments

Manuscript ID: NCOMMS-19-38798A-Z

Title: Halo-Fluorescein for Extremely Acidic Photodynamic Bacteria Inactivation

Reviewer #1 (Remarks to the Author):

In this revised manuscript, authors added another figure related to animal experiment (Fig 6) and responded to the reviewer's comment. Therefore, I think that the authors have made significant improvements to their manuscript and have satisfactorily addressed my concerns. The manuscript is acceptable for publication.

Reply: Thanks very much for your positive comment!